# A Tunnel Fire Detection Method Based on an Improved Dempster-Shafer Evidence Theory

**DOI:** 10.3390/s24196455

**Published:** 2024-10-06

**Authors:** Haiying Wang, Yuke Shi, Long Chen, Xiaofeng Zhang

**Affiliations:** 1Key Laboratory of Road Construction Technology and Equipment of Ministry of Education, Chang’an University, Xi’an 710064, China; 2023125071@chd.edu.cn (Y.S.); 2023225125@chd.edu.cn (L.C.); 2Shaanxi Transportation Holding Group Co., Ltd., Xi’an 710075, China; 18192319988@139.com

**Keywords:** tunnel, fire detection, multi-sensor data fusion, DS evidence theory, PyroSim

## Abstract

Tunnel fires are generally detected using various sensors, including measuring temperature, CO concentration, and smoke concentration. To address the ambiguity and inconsistency in multi-sensor data, this paper proposes a tunnel fire detection method based on an improved Dempster-Shafer (DS) evidence theory for multi-sensor data fusion. To solve the problem of evidence conflict in the DS theory, a two-level multi-sensor data fusion framework is adopted. The first level of fusion involves feature fusion of the same type of sensor data, removing ambiguous data to obtain characteristic data, and calculating the basic probability assignment (BPA) function through the feature interval. The second-level fusion derives basic probability numbers from the BPA, calculates the degree of evidence conflict, normalizes the BPA to obtain the relative conflict degree, and optimizes the BPA using the trust coefficient. The classical DS evidence theory is then used to integrate and obtain the probability of tunnel fire occurrence. Different heat release rates, tunnel wind speeds, and fire locations are set, forming six fire scenarios. Sensor monitoring data under each simulation condition are extracted and fused using the improved DS evidence theory. The results show that there is a 67.5%, 83.5%, 76.8%, 83%, 79.6%, and 84.1% probability of detecting fire when it occurs, respectively, and identifies fire occurrence in approximately 2.4 s, an improvement from 64.7% to 70% over traditional methods. This demonstrates the feasibility and superiority of the proposed method, highlighting its significant importance in ensuring personnel safety.

## 1. Introduction

Multi-sensors play a crucial role in the detection of tunnel fires. Due to the unique environment of tunnels, the data collected by these sensors may contain ambiguous and conflicting information, leading to issues such as false alarms and missed detections of fires [1]. Therefore, studying how to improve the accuracy of tunnel fire detection and the timeliness of fire discovery is of significant practical importance for tunnel safety.

In recent years, intelligent multi-sensors have been rapidly developed, while more high-precision fusion algorithms and models have been optimized and presented [2]. Currently, common data fusion algorithms include artificial neural networks [3], Bayesian methods [4], Kalman filtering [5], and the Dempster-Shafer (DS) evidence theory [6]. The neural network model contains input, hidden and output layers, and processes the data by mimicking biological nerves [7]. A large amount of training data are required for fusion, and the small number of tunnel fire samples makes it difficult to train effectively [8]. Bayesian methods and the DS evidence theory are often used to address multi-sensor data fusion issues [9]. However, the Bayesian method’s reliance on prior estimates can hinder its ability to accurately reflect changes in tunnel conditions, making it unsuitable for multi-sensor data fusion in tunnel fire scenarios [10]. On the other hand, the DS evidence theory is well-suited for situations where prior probabilities are unknown, allowing various sensor monitoring data to be integrated as evidence sources. However, due to the ambiguity and contradictory nature of multi-sensor data, the fusion results of DS evidence theory are paradoxical when there is a problem of highly conflicting evidence in the data [11]. Solutions to the evidence conflict problem in the DS evidence theory can be categorized into two types: altering fusion rules [12] and optimizing the BPA [13].

In terms of altering fusion rules, Xiao et al. [14] proposed a weighted combination method for conflict evidence in multi-sensor data fusion. This method adjusts the evidence weights by modifying the cosine similarity and confidence entropy to address the issue of highly conflicting evidence fusion. However, it essentially preprocesses the evidence without considering the characteristics of the evidence itself. Wang et al. [15] introduced an adaptive evidence fusion method based on the power probability distance, but it uses a single evidence relation to represent and manage conflicts and uncertainties in the Internet of Things environment, failing to effectively address the impact of unreliable evidence sources on the fusion results. Hamda et al. [16] proposed an improved evidence combination method for representing conflict and uncertainty in data fusion to improve the accuracy of decision-making, mainly relying on the Hellinger distance. Murphy [17] simply averaged the mass functions and then applied the classical DS combination rule, without considering the correlation between pieces of evidence, and assigned the same weight to all evidence sources. Tang et al. [18] proposed a new method for uncertain information fusion based on a new evidence confidence coefficient, using the evidence distance in the DS evidence theory to handle conflict and uncertain information fusion, but it introduces a single confidence function to determine the weights between evidence. Hu et al. [19] proposed a new confidence entropy method based on the internal cross-information of evidence, which adjusts the comprehensive credibility of evidence and optimizes the evidence fusion process of the DS combination rule.

In terms of optimizing the BPA, Shahpari et al. [20] proposed a pixel-based transformation uncertainty measure based on the BPA, but it has been proven in [21] that this measure does not satisfy certain properties and leads to paradoxes. Deng et al. [22] introduced an improved DS evidence theory framework based on the Hellinger distance within uncertain intervals, which is more sensitive to changes in evidence. Xiao et al. [23] suggested a multi-sensor data fusion method based on the evidence and information entropy confidence measure, using information entropy to assess the characteristics of the evidence itself. Qiao et al. [24] proposed a multi-sensor data fusion method based on the evidence theory that assigns weights according to the degree of data deviation, demonstrating high fusion accuracy. Xiao [25] introduced the concept of evidence credibility measure and designed a hybrid multi-sensor data fusion method. Song et al. [26] presented a time evidence adaptive fusion method based on a negotiation strategy, constructing an evidence set through cumulative time fusion, using the probability distance to evaluate conflicts, and adapting the fusion of time evidence according to the degree of conflict. Zhao et al. [27] proposed a new distribution distance measure method to gauge the degree of conflict between pieces of evidence, introducing a modified information amount calculation method to evaluate the role of evidence and adjust evidence credibility. Zhou et al. [28] combined indirect conflict measurement indicators with evidence information measurement indicators for data fusion, addressing issues of high conflict and poor robustness. Wang et al. [29] developed a multi-attribute fusion algorithm based on fuzzy clustering and the improved evidence theory, which uses fuzzy clustering for group measurement and then applies the improved evidence theory for advanced fusion. Although these methods incorporate relationships between pieces of evidence and the characteristics of the evidence itself, they fuse the probabilities of target attributes collected by sensors for target recognition and fault diagnosis, failing to integrate the measurement results of the sensors. Moreover, using the information entropy alone cannot comprehensively evaluate the characteristics of the evidence itself.

By comparing the two conflict resolution methods, it is evident that the methods of altering fusion rules directly eliminate the normalization step in the DS theory, redistributing the conflict using different metrics. In contrast, the methods of optimizing the BPA take into account the inherent differences between sensors in a multi-sensor system, resolving conflicts by modifying the original evidence. However, both of these methods of resolving evidence conflicts are relatively one-sided, failing to effectively combine evidence with the inherent characteristics of the evidence, resulting in a relatively low precision of data fusion.

To improve the accuracy of data fusion, this paper proposes a two-level multi-sensor data fusion framework grounded on an optimized probability assignment function. In the first level of fusion, we mainly conducted the feature integration of data from the same type of sensors. After acquiring the sensor monitoring data, we filtered out abnormal data using the Euclidean distance method and then performed the first-level fusion through the weighted average method to obtain the probability assignment function for the feature intervals. In the second level, we integrated decision features from different types of sensors, primarily optimizing the probability assignment functions obtained in the previous level. We first constructed an evidence distance matrix to calculate the degree of conflict among the evidence and normalized it to obtain the relative conflict level. Then, we calculated the trust coefficient between the pieces of evidence to further optimize the probability assignment functions. Finally, using the classical DS evidence theory, we fused the information and concluded the probability of fire in tunnels. This method aims to quickly and accurately detect tunnel fires under various fire scenarios.

## 2. The Proposed Method

This section describes the relevant knowledge of DS evidence theory and explains the conflict issues in multi-sensor data fusion for tunnel fires. Finally, it proposes methods and steps for improving the DS evidence theory.

### 2.1. Dempster-Shafer Evidence Theory

In the DS evidence theory, the frame of discernment $\Theta =\left\{\theta_{1},\theta_{2},\ldots ,\theta_{i},\ldots ,\theta_{N}\right\}$ consists of a finite set of mutually exclusive elements. Here, $2^{\Theta }$ denotes the power set of the discursive framework, which is composed of $2^{N}$ elements and is represented as follows:(1)$$\left\{\begin{array}{l} m\left(\emptyset \right)=0\\ \sum_{A\subseteq \Theta }m\left(A\right)=1\\ m\left(A\right)\geq 0 \end{array}\right.$$

Here, *m*(*A*) is referred to as the basic probability assignment (BPA) function, if $m\left(A\right)>0$, {*A*} is referred to as a proposition, and *m*(*A*) denotes the degree of belief assigned to proposition {*A*}.

In the frame of discernment Θ, ∀A⊆Θ, if the function $Bel:2^{\Theta }\to \left[0,1\right]$ satisfies:(2)$$Bel\left(A\right)=\sum_{B\subseteq A}m\left(B\right)$$

It is said that *Bel*(*A*) is the belief function of proposition {*A*}, which represents the likelihood that proposition {*A*} is true.

In the frame of discernment Θ, ∀A⊆Θ, if the function $Bel:2^{\Theta }\to \left[0,1\right]$ satisfies:(3)$$Pl\left(A\right)=\sum_{B\cap A\neq \emptyset }m\left(B\right)$$

It is said that *Pl*(*A*) is the plausibility function of proposition {*A*}, which represents the likelihood that proposition {*A*} is not false.

*Pl*(*A*) and *Bel*(*A*) are defined as the upper and lower bounds of the confidence in proposition {*A*}. The interval [*Bel*(*A*), *Pl*(*A*)] represents the degree of uncertainty of the proposition, with the relationship between *Bel*(*A*) and *Pl*(*A*) illustrated in Figure 1.

The combination rule of DS evidence theory involves merging multiple sources of evidence. Given two pieces of evidence with basic BPA *m*_1_ and *m*_2_ in the frame of discernment, the DS evidence theory fusion formula [30] for proposition {*A*} is as follows:(4)$$m\left(A\right)=⊕=\left\{\begin{array}{cc} 0 & X\cap Y=\Phi \\ \frac{\sum_{X\cap Y=A,\forall X,Y\subseteq U}m_{1}\left(X\right)\cdot m_{2}\left(Y\right)}{1-k} & X\cap Y\neq \Phi \end{array}\right.$$
where *k* serves as a pivotal conflict factor, functioning as a quantitative metric that precisely measures the degree of disagreement or disparity among diverse pieces of evidence.
(5)$$k=\sum_{X\cap Y=\Phi ,\forall X,Y\subseteq U}m_{1}\left(X\right)\cdot m_{2}\left(Y\right)$$
when *k* = 1, the DS evidence theory is unable to fuse the evidence.
(6)$$m(A)=\frac{\sum_{X\cap Y\cap Z\ldots =A}m_{1}\left(X\right)m_{2}\left(Y\right)m_{3}\left(Z\right)\ldots }{1-k}\;A\neq \Phi ,U$$
when there are three or more pieces of evidence, the combination rule of DS evidence theory is as follows:(7)$$k=\sum_{X\cap Y\cap Z\ldots =\Phi }m_{1}\left(X\right)m_{2}\left(Y\right)m_{3}\left(Z\right)\ldots$$

In the frame of discernment Θ, let *A*, *B*, and *C* represent the data from temperature sensors, CO sensors, and smoke sensors monitoring tunnel fires, respectively as follows:$$\begin{array}{l} m_{1}:m_{1}(A)=0.75,m_{1}(B)=0.15,m_{1}(C)=0.1\\ m_{2}:m_{2}(A)=0,m_{2}(B)=0.3,m_{2}(C)=0.7\\ m_{3}:m_{3}(A)=0.8,m_{3}(B)=0.15,m_{3}(C)=0.05 \end{array}$$

Using the DS evidence theory combination rule, k was calculated to be 0.98975. The data fusion results of *m*(*A*), *m*(*B*), and *m*(*C*) are shown in Table 1.

Relying solely on individual sensor data may result in temperature and smoke sensors indicating a higher likelihood of fire, while the CO sensor may indicate a normal state, leading to a paradox in data fusion based on the Dempster-Shafer (DS) evidence theory. According to the data in Table 1, under fire conditions, the value of *m*_2_(CO) is 0, which results in a post-fusion judgment of 0 for fire conditions. This causes a conflict in the DS evidence theory’s judgment.

### 2.2. The Method for Improving the Dempster-Shafer (DS) Evidence Theory

To address the conflict issue in the DS evidence theory, this paper proposes a two-level multi-sensor data fusion framework based on the optimized probability assignment function. The first level of fusion focuses on feature fusion of data from the same type of sensors; when processing data from similar sensors, a feature-level data fusion method is employed to effectively eliminate ambiguous data and enhance the accuracy of environmental monitoring by the same type of sensors, while the second level involves decision feature fusion from different types of sensors. For multi-type sensor data, decision-level fusion is used to complement multi-source data from different sensors, thereby improving the overall accuracy of the monitoring results.

The proposed framework in this paper is shown in Figure 2. The process consists of two steps, as follows:

Step 1: Primary fusion of data from sensors of the same type to obtain the BPA

The key aspect of data fusion using the Dempster-Shafer (DS) evidence theory is obtaining the BPA. Following the primary fusion process for similar sensor data, multi-sensor monitoring data are first collected, and invalid data are removed to obtain feature data. Then, a primary fusion of similar sensor data is performed. Finally, the BPA is preliminarily calculated based on the feature intervals.

Step 2: Secondary fusion of data from sensors of multiple types to obtain tunnel conditions information

The improved DS evidence theory addresses evidence conflict issues and yields the final fusion results of multi-sensor data. According to the secondary fusion process for multi-sensor data, the basic probability numbers are first derived from the BPA. Then, an evidence distance matrix is constructed to calculate the degree of conflict between pieces of evidence, and normalization is performed to obtain the relative conflict degree. Subsequently, the trust coefficients between pieces of evidence are calculated to further optimize the BPA. Finally, the classical DS evidence theory is used to fuse the evidence and determine the tunnel conditions.

#### 2.2.1. Primary Fusion of Data from the Same Type of Sensors

The process for obtaining the improved DS evidence theory BPA function for the fusion of data from the same type of sensors involves the following steps:

(1)Similar data screening based on the Euclidean distance

To exclude anomalous data when calculating the BPA, this paper proposes a similar data screening algorithm based on the Euclidean distance. The primary method involves measuring the similarity between data points by calculating the Euclidean distance between similar data. A smaller distance indicates a higher degree of similarity and greater data authenticity, while a larger distance indicates a lower degree of similarity and lesser data authenticity. Therefore, by calculating the pairwise distances between all similar data points and setting an appropriate threshold, anomalies can be identified and excluded [31].

Let the number of sensors be *n*, and the distance between the collected data $s_{i}=\left\{s_{i}|i=1,2,\ldots ,n\right\}$, then the distance $d\left(s_{i}\right)$ [32] between $s_{i}$ and other similar data excluding $s_{i}$ can be expressed as follows:(8)$$\Delta d(i)=d\left(s_{i}\right)-d_{m}$$
(9)$$d\left(s_{i}\right)={\left[\frac{1}{n}\sum_{j=1}^{n}{\left(s_{i}-s_{j}\right)}^{2}\right]}^{\frac{1}{2}},i\neq j,i=1,2,\ldots ,n$$

The distance $d_{y}$ of anomalous data is greater than the distance $d_{z}$ of normal data. Therefore, the difference Δd(i) between the distances of anomalous and normal data can be used to identify anomalous data. In the equation, $d_{m}$ represents the median of $d\left(s_{i}\right)$.

The magnitude of Δd(i) can be eliminated by the following equation:(10)$$dw\left(s_{i}\right)=\frac{\Delta d(i)}{s_{m}}$$
where $s_{m}$ represents the median of the set of sensor data of the same type $s_{i}$.

When $dw\left(s_{i}\right)\geq 0.02$, the sensor is considered to have a large error and should be rejected.

(2)Primary fusion of homogeneous sensor data using a weighted averaging method

If the weights are the same between sensors of the same type, then it is known that
(11)$$\bar{x}=\frac{1}{n}\sum_{i=1}^{n}x_{i}$$

We defined $\bar{x}$ as the fused value of data from the same type of sensors and $L(\bar{x})$ as the distance from $\bar{x}$ to other sensors. When $L(\bar{x})$ reached its minimum value, $\bar{x}$ was closest to the local multi-sensor monitoring result, as follows:(12)$$L(\bar{x})={\sum_{i=1}^{n}\left(\bar{x}-x_{i}\right)}^{2}$$

Derived that
(13)$$L(\bar{x})=n{\bar{x}}^{2}+\sum_{i=1}^{n}x_{i}^{2}-2\bar{x}\sum_{i=1}^{n}x_{i}$$

Derivation on the left and right sides yields that
(14)$$\frac{dL(\bar{x})}{d\bar{x}}=2n\bar{x}-2\sum_{i=1}^{n}x_{i}$$
when $\bar{x}=\frac{1}{n}\sum_{i=1}^{n}x_{i}$, the minimum value of $L(\bar{x})$ is obtained, which corresponds to the average of the data from all sensors.

(3)Acquisition of BPA functions based on feature intervals

To monitor tunnel fire conditions in real-time, it is essential to collect three key types of data: carbon monoxide (CO) concentration, smoke concentration, and temperature. The identification framework is $\Theta =\{A_{1},A_{2},\ldots ,A_{n-1}A_{n}\}$, which categorizes the tunnel fire status into three levels: {Normal conditions, Warning conditions, and Fire conditions}.

Interval T represents the range within which the identification framework Θ is situated. This interval divides the *n* objects within the identification framework into *n* characteristic intervals $T_{i}$, i=1,2,…,n, which describes the range $\left[T_{iL},T_{iR}\right]$ where the identification object $A_{i}$ is located. The midpoint $T_{iM}$ of each interval is set as the characteristic value for that feature interval, as follows:(15)$$T_{iM}=\frac{T_{iL}-T_{iR}}{2}\;i=1,2,\ldots ,n$$
where $T_{iL}$ represents the left boundary of the characteristic interval $T_{i}$, while $T_{iR}$ represents the right boundary of the characteristic interval $T_{i}$.

Let $l{\left(j\right)}_{i}$ represent the distance between the sensor’s measured value a(j) and each characteristic value, as follows:(16)$$l{\left(j\right)}_{i}=\left|a(j)-T_{iM}\right|\;i=1,2,\ldots ,n$$

Dividing $l{\left(j\right)}_{i}$ by the length of each characteristic interval yields the dimensionless distance $lw{\left(j\right)}_{i}$ between the sensor’s measured value and each interval’s characteristic value, as follows:(17)$$lw{\left(j\right)}_{i}=\frac{1}{T_{iL}-T_{iR}}l{\left(j\right)}_{i}\;i=1,2,\ldots ,n$$

Taking the reciprocal of $lw{\left(j\right)}_{i}$ and normalizing it yields the probability assignment function mass(j) and the basic probability number $mass{\left(j\right)}_{i}$ for the measured value a(j), as follows:(18)$$mass{\left(j\right)}_{i}=\frac{1/(lw{\left(j\right)}_{i}+C_{0})}{\sum_{k=1}^{n}\left[1/(lw{\left(j\right)}_{k}+C_{0})\right]}\;i=1,2,\ldots ,n$$
where $C_{0}$ is a constant.

According to Equation (18), the closer the sensor’s measured value is to the characteristic value $T_{iM}$ of the interval, the larger $mass{\left(j\right)}_{i}$ becomes, indicating that a(j) is closer to the identification object $A_{i}$. However, during a tunnel fire, some sensor measurements may exceed the right boundary $T_{nR}$ of the characteristic interval $T_{n}$, causing $lw{\left(j\right)}_{i}$ to become too large and, consequently, $mass{\left(j\right)}_{i}$ to decrease, which moves a(j) further away from the identification object $A_{i}$. Therefore, when calculating $mass{\left(j\right)}_{i}$, if a(j) is too large, it should be replaced with $a{\left(j\right)}^{\prime }$. Based on the characteristic interval $T_{i}$, this adjustment ensures that a(j) remains close to the corresponding characteristic value $T_{iM}$.
(19)$$a{\left(j\right)}^{\prime }=T_{iM}+C_{1}$$
where $C_{1}$ is a constant.

#### 2.2.2. Secondary Fusion of Multi-Type Sensor Data

The improved DS evidence theory for multi-type sensor data fusion is applied to a secondary fusion process.

First, let the identification framework be $\Theta =\{A_{1},A_{2},A_{3},\ldots ,A_{n}\}$, where $A_{n}$ represents the *n*-th condition of tunnel fire, and $E=\{E_{1},E_{2},E_{3},\ldots ,E_{m}\}$ denotes the evidence set, with $E_{m}$ being the m-Th piece of evidence related to the tunnel fire. The definition of the distance $d_{ij}$ between evidence $mass_{j}$ and $mass_{i}$ is given by the following:(20)$$d_{ij}=e^{1-{<E}_{i},E_{j}>}i,j=1,2,\ldots ,m$$
where
(21)$${<E}_{i},E_{j}>=\frac{\sum_{s=1}^{m}mass_{i}\left(A_{s}\right)mass_{j}\left(A_{s}\right)}{\sqrt{\sum_{s=1}^{m}mas{s_{i}}^{2}\left(A_{s}\right)}\sqrt{\sum_{s=1}^{m}mas{s_{j}}^{2}\left(A_{s}\right)}}$$

As $d_{ij}$ approaches 1, it indicates a higher degree of mutual support between the pieces of evidence. Conversely, as $d_{ij}$ approaches e, it signifies a higher level of conflict between the pieces of evidence. Based on this, an evidence distance matrix D can be constructed:(22)$$D=\left[\begin{array}{cccc} 1 & d_{12} & \ldots & d_{1m}\\ d_{21} & 1 & \ldots & d_{2m}\\ \vdots & \vdots & \ddots & \vdots \\ d_{m1} & d_{m2} & \ldots & d_{mm} \end{array}\right]$$

Defining $con(E_{i})$ as the level of conflict between evidence gives the following:(23)$$con\left(E_{i}\right)=\sum_{i=1,i\neq j}^{m}d_{ij}$$

Normalizing the evidence conflict degree to obtain the relative conflict degree of the evidence $in(E_{i})$ gives the following [33]:(24)$$in\left(E_{i}\right)=\frac{con\left(E_{i}\right)}{\sum_{j=1}^{m}con\left(E_{j}\right)}$$

Let $\lambda (E_{i})$ be the trust coefficient of the evidence, representing the importance of evidence $E_{i}$ and its influence on the fusion result. The definition of $\lambda (E_{i})$ is as follows:(25)$$\lambda \left(E_{i}\right)=\left[1-in\left(E_{i}\right)\right]e^{-in\left(E_{i}\right)}$$

Let $m_{i}$ be the original probability assignment function. The optimized probability assignment function $mass_{i}^{*}$ is as follows:(26)$$mass_{i}^{*}(A_{j})=\lambda \left(E_{i}\right)\cdot mass_{i}(A_{j})$$
(27)$$mass_{i}^{*}(X)=1-\lambda \left(E_{i}\right)$$
where $mass_{i}^{*}(X)$ denotes uncertainty.

In summary, the specific steps for improving the DS evidence theory for the secondary fusion of multi-sensor data in tunnels are as follows:

Firstly, an evidence distance matrix D is constructed to calculate the degree of conflict between the evidence and normalized BPA to obtain the relative degree of conflict.

Secondly, the trust coefficients are calculated between the evidence to optimize the BPA.

Finally, the evidence is fused using the classical DS theory of evidence.

## 3. Numerical Examples and Simulation Verification

This section evaluates and simulates the proposed method based on the improved DS evidence theory to verify its feasibility and effectiveness.

### 3.1. Improved Dempster-Shafer (DS) Evidence Theory

To validate the effectiveness of the proposed improvement, sensor monitoring data from Section 2.1 are used to compare the proposed improved algorithm with the methods developed by Dempster-Shafer [34], Yager [35], Sun [36], and Murphy [17]. The fusion results are shown in Table 2 and the comparison of the fusion results is illustrated in Figure 3.

From Table 1, it can be observed that under fire condition A, when *m*_1_ = 0.75, *m*_2_ = 0, and *m*_3_ = 0.8, the BPA *m*(*A*) for Yager’s method is 0, which results in a paradox. In contrast, from Table 2 the BPA for fire conditions in this paper is 0.50247, indicating the detection of a fire. When dealing with different types of conflicting evidence, classical DS and Yager’s fusion methods lead to contradictions with the actual facts. Although the method proposed by Sun et al. points to the correct result, it exhibits a higher degree of uncertainty. Therefore, traditional DS fusion rules may fail or mismatch the actual tunnel fire conditions when confronted with conflicting evidence. This paper addresses the problem of evidence conflict by reducing the proportion of conflicting evidence, gradually decreasing the number of conflicting evidence, and increasing the amount of valid evidence, thus minimizing the uncertainty interval and enhancing reliability. The improved DS evidence theory fusion algorithm proposed in this paper provides more accurate results in handling evidence and conflicting evidence, demonstrating the rationality and effectiveness of the proposed method.

### 3.2. The Proposed Holistic Approach

To further demonstrate the feasibility and effectiveness of the data fusion method, this subsection applies the proposed improved DS evidence theory data fusion method to tunnel fire detection.

In tunnel fire detection, temperature sensors, CO sensors, and smoke sensors are commonly used to monitor fires. The PyroSim (2010) software, based on the Fire Dynamics Simulator (FDS) and Computational Fluid Dynamics, can simulate large-scale slow-moving vortices and accurately obtain critical parameters such as fire heat release, fire smoke, CO concentration, and temperature.

The fire simulation process using PyroSim is illustrated in Figure 4. First, a comprehensive tunnel geometric model is created, including elements such as lining, vehicles, and ventilation systems, to accurately describe the internal structure of the tunnel. Next, multiple sensors, including temperature, CO, and smoke sensors, are installed to enable real-time monitoring of the fire scene. The heat release rate is determined based on the actual fire conditions. Subsequently, a tunnel fire simulation model is established, and simulation parameters such as temperature and velocity are set to realistically simulate the fire situation. Finally, FDS is run to perform the fire simulation and obtain multi-sensor monitoring data.

#### 3.2.1. Simulation Model

(1)Establishing the tunnel geometric model

The tunnel geometric model is constructed with the following specifications: a height of 8.5 m, a width of 14.5 m, and a length of 600 m. The tunnel is lined with 31 concrete walls forming the lining and surrounding rock structure. The wall material has a density of 2280.0 kg/m^3^, a specific heat capacity of 1.04 kJ/(kg·K), and a thermal conductivity of 1.8 W/(m·K). In consideration of practical scenarios, the dimensions of the vehicles are set as follows: width 2.4 m, height 2.5 m, and length 7 m. The fire materials carried by large trucks are wood and plastic. In the tunnel model, a blue square on the top represents the jet fan, which has an outlet area of 1.21 m^2^ and a wind speed of 27.9 m/s. These parameters are used to establish the tunnel geometric model, with the specific structure illustrated in Figure 5.

(2)Installation of fire monitoring equipment

A 3 × 3 sensor matrix is arranged on the tunnel ceiling, with each matrix containing three temperature sensors, three CO sensors, and three smoke sensors. The distance between adjacent sensor matrices is 10 m. The specific arrangement of the sensors is illustrated in Figure 6.

The tunnel fire monitoring sensor cross-section is divided into six types of sections. The location of each section and sensors is designed primarily in accordance with the reference standard [37]. Sections S1 and S3 represent examples of smoke monitoring sections. Section S1 is located near the tunnel entrance fan, and section S3 is in the middle of the tunnel, with a distance of 75 m between adjacent smoke monitoring sections. Sections S2 and S6 are wind speed monitoring sections, with section S2 located 175 m from the tunnel entrance and section S6 positioned 225 m from the tunnel exit. Section S4 represents an example of a fire monitoring section, while Section S5 is a convergence node section located in the middle of the tunnel for transmitting multi-sensor monitoring data.

The locations of sensors for smoke and wind speed are depicted in Figure 7. Smoke produced by a fire poses the greatest threat to people evacuating, making it essential to monitor the height of the smoke layer in real-time. When the smoke layer is 3 m above the ground, it indicates that it is about to threaten those escaping. When it reaches 2 m above the ground, it has already entered the evacuation space. Therefore, smoke sensors are positioned at heights of 2 m and 3 m above the ground. Wind speed sensors are placed on the side of the tunnel at a height of 3 m.

The sensor locations for fire monitoring cross-sections are illustrated in Figure 8. When a fire occurs in the tunnel, smoke, heat, and CO typically accumulate at the tunnel ceiling before flowing along the ceiling in the direction of the wind. Therefore, placing the fire monitoring nodes at the top of the tunnel allows for quicker detection of the tunnel’s scenario. To accurately determine the location of a fire, the spacing between fire monitoring nodes should not be too large. Additionally, given the high temperatures at the tunnel ceiling during a fire, any sensor failure should be compensated by adjacent sensors. Considering these factors, the interval between fire monitoring cross-sections is set to 10 m [38].

(3)Establishing the fire model

According to The National Fire Protection Association of the United States of America in the ‘tunnel and underground space standards’ for different models of fire power in the event of a fire, this paper sets the size of the model of the corresponding heat release rate, respectively, 20 MW and 5 MW [39]. According to the standard [40], the wind speeds at the middle, quarter, and entrance of the tunnel were set at 2.5 m/s, 5 m/s, and 8 m/s, respectively. The air pressure was set at 94.5 KPa, accompanied by the initial tunnel temperature which was set at 20 °C. Additionally, the initial CO concentration in the tunnel was set at 42 cm^3^/m^3^, and the initial smoke concentration in the tunnel k was set at 0.004 m^−1^. As depicted in Figure 9, a strategic layout was adopted for the fire simulation, with fire point 1 positioned at the tunnel’s midpoint, representing the primary fire scenario. Conversely, fire points 2 and 3 were situated in close proximity to the wind turbines, serving as control scenarios. Given the inherent challenges associated with remote fire detection, it was determined that the distance between the inception point and the downstream sensor matrix of a fire should be maintained at 7.5 m, ensuring optimal monitoring capabilities within practical constraints.

Based on the established fire conditions, with other fire conditions remaining consistent, six tunnel fire scenarios are set as shown in Table 3.

#### 3.2.2. Tunnel Fire Simulation

Based on the tunnel fire scenarios set in Section 3.2.1, simulations are conducted in PyroSim. Using the fire scenario LZ-R5-S2.5 as an example, the multi-sensor monitoring data for tunnel fires are obtained. The monitoring data for temperature sensors, smoke sensors, and CO sensors in the fire monitoring nodes are shown in Table 4.

#### 3.2.3. Tunnel Fire Simulation Data Fusion

Based on the standard [40] and relevant design standards for tunnel engineering, the characteristic intervals for temperature, smoke concentration, and CO concentration are categorized as shown in Table 5.

Combining Table 5, we conducted the first-level fusion of the sensor-monitored data under the fire scenario LZ-R5-S2.5. Using Equations (18) and (19), we calculated the probability assignment function, as shown in Table 6.

The probability assignment function is optimized using Equations (26) and (27), and the probability assignment function for uncertainty is calculated. Combining the optimized probability assignment function with Equation (6), the probabilities of Fire condition, Fire warning condition, Normal operating condition, and Uncertainty under various tunnel condition are determined, as shown in Table 7.

For the different working conditions mentioned above, the same method was applied, and the resulting probability curves for each state are shown in Figure 10.

As shown in Figure 10, for the fire scenario LZ-R5-S2.5, the system detected that the temperature and smoke concentration exceeded the tunnel design upper limits at 4.2 s, and simultaneously detected the fire. Within the first 10 s of the fire, the highest probability of fire occurrence detected by the system was 67.5%. In the fire scenario LZ-R20-S2.5, the temperature and smoke concentration exceeded the design upper limits at 3 s, and the fire was detected at 3.6 s. In the first 10 s, the system recorded a maximum fire occurrence probability of 83.5%. For the fire scenario LQ-R5-S5, the system detected the temperature and smoke concentration exceeding the design upper limits at 3.6 s, and fire detection occurred at 4 s. The maximum fire occurrence probability during the first 10 s was 76.8%. In the fire scenario LQ-R20-S5, the temperature and smoke concentration exceeded the design upper limits at 2.4 s, and the fire was detected simultaneously. The highest fire occurrence probability in the first 10 s was 83%. For the fire scenario LEN-R5-S8, the system detected the temperature and smoke concentration exceeding the design upper limits at 2.4 s, and fire detection occurred at 3 s. The maximum probability of fire occurrence in the first 10 s was 79.6%. Finally, for the fire scenario LEN-R20-S8, the temperature and smoke concentration exceeded the design upper limits at 2.4 s, and fire detection occurred simultaneously. The system recorded the highest probability of fire occurrence at 84.1% within the first 10 s.

The fire occurrence probabilities under different fire scenarios are analyzed, and the resulting curves for the six tunnel fire scenarios are shown in Figure 11.

From the multi-sensor monitoring data and fusion results for different fire scenarios, it is evident that the higher the heat release rate and wind speed, the easier it is to detect a tunnel fire. Due to varying tunnel wind speeds and heat release rates, the probability of fire occurrence under specific conditions differs. In the fire scenario LM-R5-S2.5, the low heat release rate and low wind speed make fire detection most challenging, resulting in the lowest probability of fire occurrence. Conversely, in the fire scenario LZ-R20-S2.5, with the same location but higher heat release rate and wind speed, the probability of fire occurrence significantly increases.

As seen in Figure 11, the analysis of the six different fire scenarios shows that in all scenarios, the system detects the fire within 5 s. This demonstrates that the improved DS evidence theory-based multi-sensor data fusion algorithm enables the system to detect fire occurrences promptly. The system determines the occurrence of a tunnel fire when two types of monitoring data exceed the tunnel’s design thresholds. This approach allows for scenarios where one type of sensor might not detect abnormal data while preventing false alarms from single-sensor abnormalities. These results further prove the robustness of the improved DS evidence theory-based multi-sensor data fusion algorithm.

The comparison results of the improved DS evidence theory multi-sensor fusion algorithm with those of Sun [36] and the original methods are shown in Figure 12.

Figure 12 shows the fire occurrence probabilities under the same scenario, LZ-R20-S2.5, using various methods. It can be observed that in the first 2 s before the fire, the changes in the tunnel environment were minimal, and the differences between the three algorithms were negligible. However, as the heat release rate of the fire increased, the temperature and smoke concentration inside the tunnel began to rise. The method proposed in this paper was more sensitive to the data, enabling quicker detection of the tunnel fire and calculating a higher fire occurrence probability under this scenario. Compared to other methods, the proposed method improved fire detection accuracy from 5% to 10.2% and reduced the fire detection time to approximately 2.4 s, which is an improvement from 60% to 70%. This demonstrates the feasibility and effectiveness of the proposed method in multi-sensor data fusion for fire monitoring.

## 4. Conclusions

This paper proposes a multi-sensor data fusion framework based on the improved DS evidence theory, which effectively monitors the environment and fire conditions within a tunnel, significantly enhancing the accuracy and efficiency of monitoring.

A multi-sensor data fusion algorithm is proposed based on an improved DS evidence theory, employing a two-level fusion framework. Initially, the data collected by different sensors is screened to eliminate inaccurate data. Subsequently, primary fusion is performed on data from sensors of the same type. Next, the BPA functions for fire, fire warning, and normal operating conditions are extracted from the results of the primary fusion. The evidence conflicts are treated as a manifestation of uncertainty, and the BPA functions are optimized. Finally, the optimized evidence is fused using the DS evidence theory to achieve an accurate assessment of the tunnel’s operational status.

To address the conflict issue in the DS evidence theory, an improved DS evidence theory fusion algorithm is proposed. Comparisons with other data fusion algorithms show that the proposed method achieves a smaller uncertainty interval (*m*(*X*) = 0.19679). By making full use of tunnel fire data information, the fusion results exhibit higher credibility.

The proposed multi-sensor data fusion algorithm was validated using multi-sensor monitoring data. The results indicate that the method consistently achieves a fire detection probability of no less than 65% across six different fire scenarios. Compared to other research methods, the proposed algorithm offers faster analysis speeds and makes more comprehensive use of tunnel environment information. This demonstrates its potential for practical application in tunnel safety monitoring systems and suggests that it could enhance the early warning capabilities for tunnel fires.

## Figures and Tables

**Figure 1 sensors-24-06455-f001:**
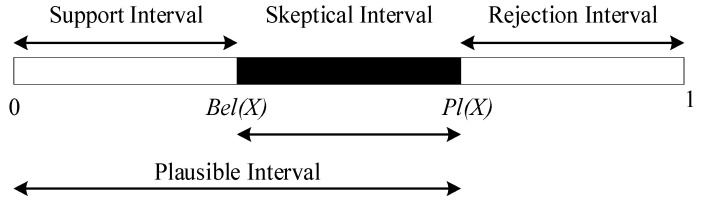
DS evidence theory confidence interval.

**Figure 2 sensors-24-06455-f002:**
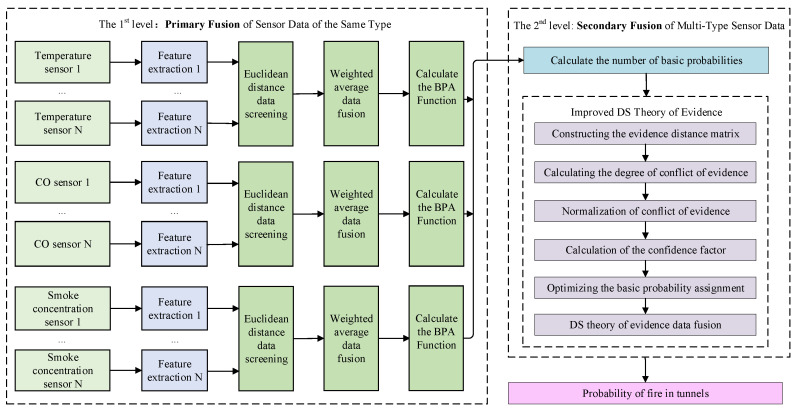
The proposed two-level multi-sensor data fusion framework based on the improved DS evidence theory.

**Figure 3 sensors-24-06455-f003:**
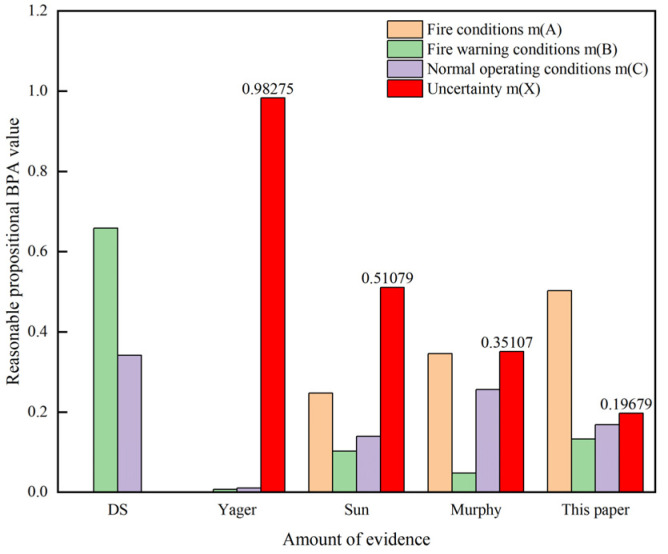
Comparison of data fusion results for five algorithms.

**Figure 4 sensors-24-06455-f004:**
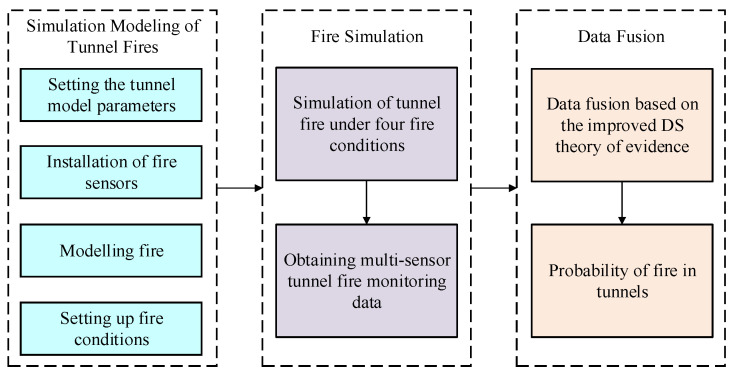
Tunnel fire simulation process.

**Figure 5 sensors-24-06455-f005:**
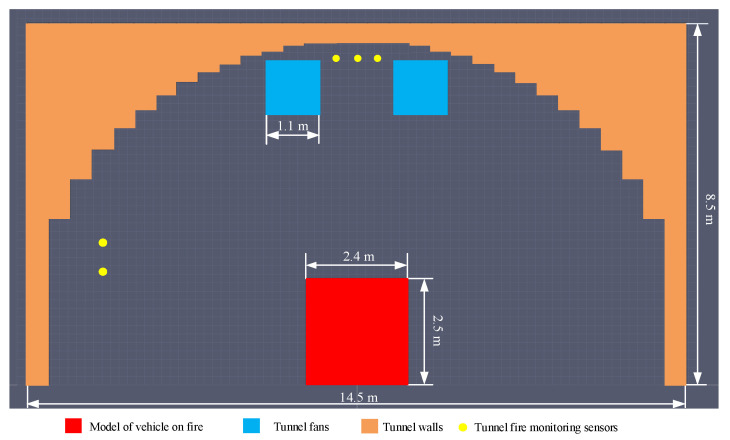
Tunnel geometric model.

**Figure 6 sensors-24-06455-f006:**
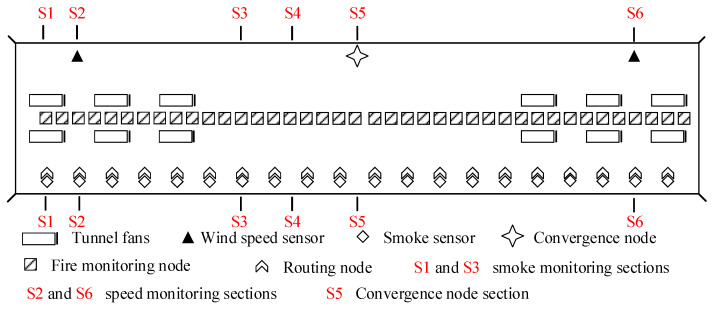
Tunnel fire monitoring sensor layout.

**Figure 7 sensors-24-06455-f007:**
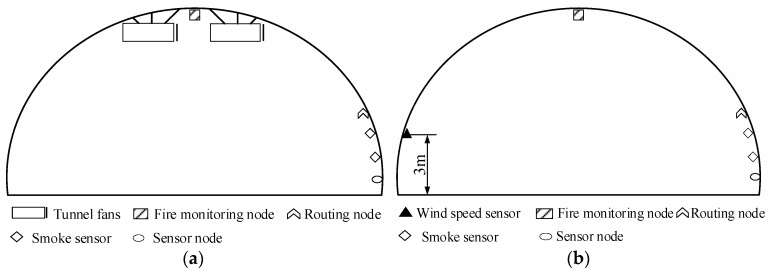
Layout of tunnel smoke and wind speed monitoring cross-section. (**a**) Tunnel smoke monitoring cross-section S1; (**b**) tunnel smoke monitoring cross-section S3; (**c**) tunnel wind speed monitoring cross-section S2; (**d**) local cross-section monitoring of tunnel wind speed S6.

**Figure 8 sensors-24-06455-f008:**
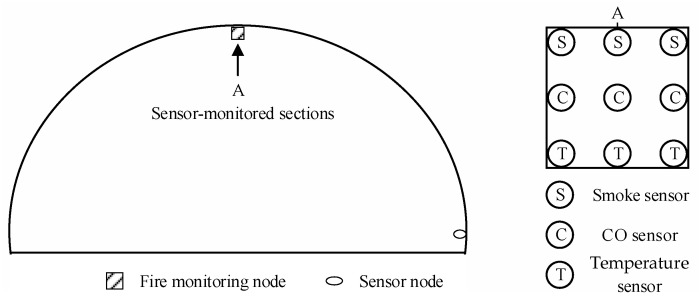
Layout of the tunnel fire monitoring cross-section.

**Figure 9 sensors-24-06455-f009:**
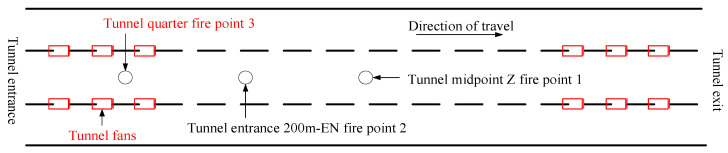
Tunnel ignition point locations.

**Figure 10 sensors-24-06455-f010:**
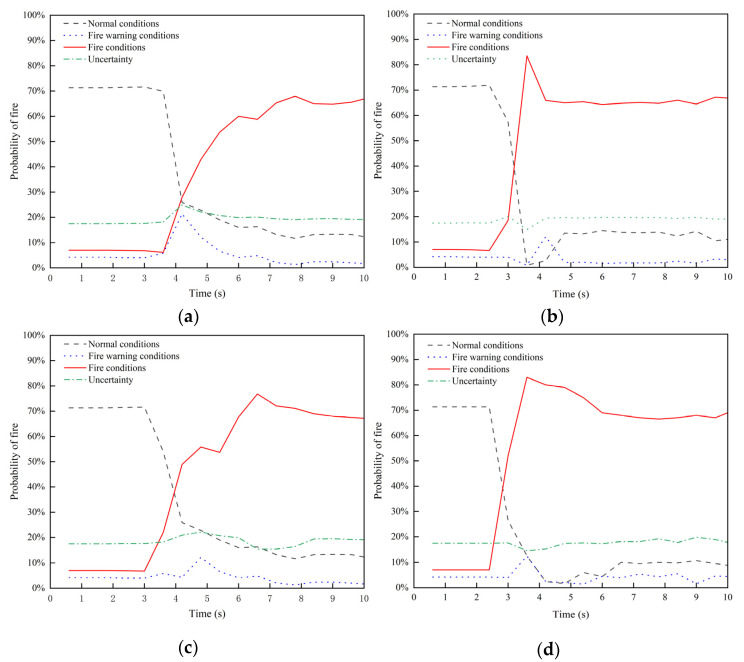
Probability curves of tunnel fire occurrence under different fire scenarios. (**a**) LZ-R5-S2.5; (**b**) LZ-R20-S2.5; (**c**) LQ-R5-S5; (**d**) LQ-R20-S5; (**e**) LEN-R5-S8; (**f**) LEN-R20-S8.

**Figure 11 sensors-24-06455-f011:**
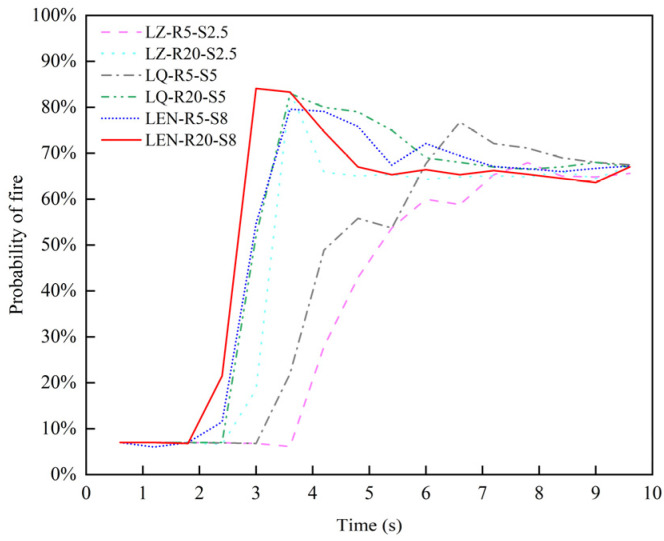
Comparison of tunnel fire occurrence probability curves under six fire scenarios.

**Figure 12 sensors-24-06455-f012:**
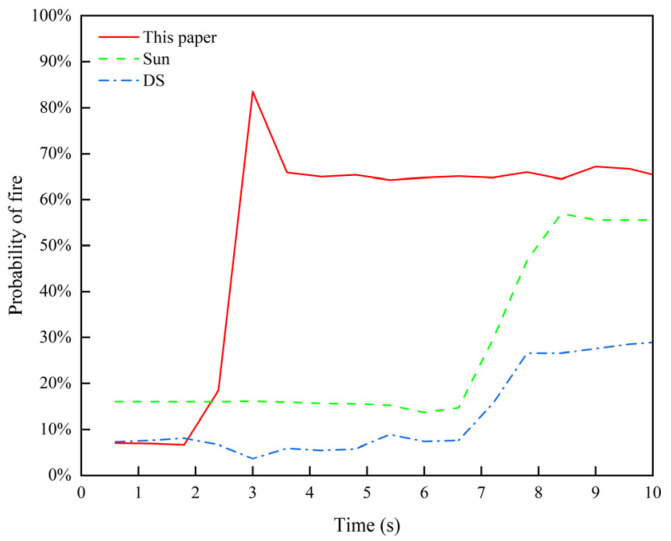
Comparison of fire state prediction curves.

**Table 1 sensors-24-06455-t001:** DS theory of evidence data fusion results.

Actual Tunnel Conditions	Multi-Sensor Fire BPA			Data Fusion Fire Probability Decision
	Temperature *m*_1_	CO *m*_2_	Smoke Concentration *m*_3_	Data Fusion Results
Fire condition A	0.75	0	0.8	0
Fire warning condition B	0.15	0.3	0.15	0.6585
Normal operating condition C	0.1	0.7	0.05	0.3415

**Table 2 sensors-24-06455-t002:** Data fusion results of five algorithms.

	Fire Condition *m*(*A*)	Fire Warning Condition *m*(*B*)	Normal Operating Condition *m*(*C*)	Uncertainty *m*(*X*)
DS	0	0.65850	0.34150	
Yager [35]	0	0.00675	0.01050	0.98275
Sun [36]	0.24746	0.10254	0.13921	0.51079
Murphy [17]	0.34536	0.04751	0.25606	0.35107
This paper	0.50247	0.13253	0.16821	0.19679

**Table 3 sensors-24-06455-t003:** Tunnel fire scenarios.

Fire Scenarios	Ignition Point L	Heat Release Rate (MW)	Wind Speed (m/s)
LZ-R5-S2.5	Tunnel Midpoint-Z	5	2.5
LZ-R20-S2.5	Tunnel Midpoint-Z	20	2.5
LQ-R5-S5	Tunnel Quarter-Q	5	5
LQ-R20-S5	Tunnel Quarter-Q	20	5
LEN-R5-S8	Tunnel Entrance 200 m-EN	5	8
LEN-R20-S8	Tunnel Entrance 200 m-EN	20	8

**Table 4 sensors-24-06455-t004:** Multi-sensor monitoring data for the tunnel fire scenario LZ-R5-S2.5.

Time (s)	Temperature (°C)			Smoke Concentration (m^−1^)			CO Concentration (cm^3^/m^3^)		
	1	2	3	Gas 1	Gas 2	Gas 3	CO_1	CO_2	CO_3
0.6	20	20	20	0.0040	0.0040	0.0040	42	42	42
1.2	20.01	20.01	20.01	0.0040	0.0040	0.0040	42	42	42
1.8	20.04	20.04	20.04	0.0040	0.0040	0.0040	42	42	42
2.4	20.23	20.23	20.23	0.0040	0.0040	0.0040	42	42	42
3	50	20.55	20.55	0.0040	0.0040	0.0040	42	42	42
3.6	28.95	28.99	28.9	0.0070	0.0070	0.0070	51	51	51
4.2	46.22	46.7	46.22	0.0707	0.0706	0.0708	48	48	48
4.8	54.75	55.73	54.75	0.0597	0.0595	0.0599	43	43	43
5.4	62.03	63.39	62.03	0.0588	0.0587	0.0589	44	44	44
6	67.59	68.72	67.59	0.0591	0.0591	0.0591	44	44	44
6.6	66.65	67.46	66.65	0.0624	0.0624	0.0625	43	43	43
7.2	72.59	73.28	72.59	0.0394	0.0396	0.0392	44	44	44
7.8	78.1	78.77	78.1	0.0569	0.0587	0.0551	43	43	43
8.4	82.01	82.82	75	0.0604	0.0598	0.0610	42	42	42
9	82.05	82.7	82.05	0.0662	0.0655	0.0669	42	42	42
9.6	79.91	80.46	79.91	0.0719	0.0724	0.0715	42	42	42
10.2	79.08	79.77	79.08	0.0525	0.0539	0.0512	43	42	44
10.8	75.8	77.15	75.8	0.0536	0.0518	0.0554	43	42	43

Note: A 600-s fire simulation was executed within PyroSim, and Table 4 encapsulates the crucial fire simulation data extracted from the initial 10.8 s of this simulation.

**Table 5 sensors-24-06455-t005:** Feature interval classification.

State Interval	Normal Operating Condition	Fire Warning Conditions	Fire Conditions
Temperature range (°C)	0~40	40~55	55~100
Smoke concentration range (m^−1^)	0.0000~0.0075	0.0075~0.0120	0.0120~0.0750
CO concentration interval (cm^3^/m^3^)	0.00~107.50	107.50~150.00	150.00~350.00

**Table 6 sensors-24-06455-t006:** Probability assignment functions for the fire scenario LZ-R5-S2.5.

Time (s)	Temperature *m*_1_			Smoke Concentration *m*_2_			CO Concentration *m*_3_		
	Normal Operating Condition	Fire Warning Condition	Fire Condition	Normal Operating Condition	Fire Warning Condition	Fire Condition	Normal Operating Condition	Fire Warning Condition	Fire Condition
0.6	0.60	0.17	0.24	0.91	0.03	0.06	0.85	0.05	0.10
1.2	0.60	0.17	0.24	0.91	0.03	0.06	0.85	0.05	0.10
1.8	0.60	0.17	0.24	0.91	0.03	0.06	0.85	0.05	0.10
2.4	0.60	0.16	0.23	0.91	0.03	0.06	0.85	0.05	0.10
3	0.61	0.16	0.23	0.91	0.03	0.06	0.85	0.05	0.10
3.6	0.90	0.05	0.05	0.42	0.29	0.29	0.95	0.02	0.03
4.2	0.08	0.82	0.10	0.05	0.03	0.92	0.91	0.03	0.06
4.8	0.17	0.41	0.42	0.04	0.02	0.94	0.87	0.05	0.09
5.4	0.14	0.22	0.64	0.04	0.02	0.94	0.87	0.04	0.08
6	0.09	0.13	0.78	0.04	0.02	0.94	0.87	0.04	0.09
6.6	0.10	0.14	0.76	0.04	0.03	0.93	0.86	0.05	0.09
7.2	0.05	0.06	0.89	0.02	0.01	0.97	0.87	0.04	0.08
7.8	0.01	0.01	0.98	0.03	0.02	0.95	0.86	0.05	0.09
8.4	0.04	0.05	0.91	0.04	0.03	0.94	0.85	0.05	0.10
9	0.04	0.05	0.91	0.04	0.03	0.93	0.85	0.05	0.10
9.6	0.03	0.03	0.95	0.05	0.03	0.92	0.85	0.05	0.10
10.2	0.02	0.02	0.96	0.02	0.02	0.96	0.87	0.05	0.09
10.8	0.02	0.02	0.97	0.03	0.02	0.96	0.86	0.05	0.09

**Table 7 sensors-24-06455-t007:** Tunnel states computed using the improved DS evidence theory for the fire scenario LZ-R5-S2.5.

Time (s)	Normal Operating Condition	Fire Warning Condition	Fire Condition	Uncertainty
0.6	71.30%	4.20%	7.00%	17.50%
1.2	71.30%	4.20%	7.00%	17.50%
1.8	71.30%	4.20%	7.00%	17.50%
2.4	71.50%	4.00%	6.90%	17.60%
3	71.60%	4.00%	6.80%	17.60%
3.6	69.90%	5.80%	6.10%	18.20%
4.2	25.90%	21.30%	27.90%	24.90%
4.8	22.80%	12.20%	42.90%	22.10%
5.4	19.00%	6.60%	53.70%	20.70%
6	16.00%	4.10%	60.00%	19.90%
6.6	16.30%	4.80%	58.80%	20.10%
7.2	13.20%	2.10%	65.30%	19.40%
7.8	11.70%	1.30%	67.90%	19.10%
8.4	13.20%	2.40%	65.00%	19.40%
9	13.30%	2.40%	64.80%	19.50%
9.6	13.20%	2.00%	65.60%	19.20%
10.2	11.90%	1.50%	67.50%	19.10%
10.8	12.00%	1.50%	67.50%	19.00%

## Data Availability

The original contributions presented in the study are included in the article, further inquiries can be directed to the corresponding author.

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
