# Peer review of "A Tunnel Fire Detection Method Based on an Improved Dempster-Shafer Evidence Theory"

_sensors, 2024, doi:10.3390/s24196455_

Round 1

Reviewer 1 Report

Comments and Suggestions for Authors

This paper proposes an improved Dempster-Shafer (DS) evidence theory for tunnel fire detection using multi-sensor data fusion. To address issues of ambiguity and evidence conflict in sensor data, a two-level fusion framework is employed. The first level fuses data from similar sensors to remove ambiguity and calculate the basic probability assignment (BPA). The second level refines the BPA by considering evidence conflict and optimizing it with a trust coefficient. The method successfully detects fires with probabilities of 67.5%, 71.0%, 82.8%, and 83.5% across four scenarios, and improves detection speed by 64.7% to 70% over traditional methods. This approach enhances safety by quickly identifying fire occurrences. In my opinion, this manuscript can be published on the journal of Sensors after the following questions being addressed.

1.      Once you mentioned thatDespite its advantages in handling uncertainty, DS evidence theory can produce paradoxical results when there is a high degree of evidence conflict.” Can you make a detailed analysis of the main causes of evidence conflicts and why data of evidence conflicts are produced?

2.      Among all the methods to solve the conflict of DS evidence, what is the most important innovation in data fusion in this paper?

3.      In Fig. 6, whether S1 and S2 in the schematic diagram represent sensor arrays? And whether the current tunnel sensor distribution map can be described in more detail because there are many symbols at present.

4.      As mentioned in the paper, the author simulated a 600m long tunnel and could you explain in detail the considerations for the arrangement of array sensors in the tunnel, why they are arranged in this way, and whether there are any advantages in spatial planning and efficiency?

5.      In Fig. 6 and Fig. 7, the author mentioned two kinds of fans, tunnel fans, and jet fan, are these two mean the same thing? If so, maybe using the same icon can be clearer and intelligible, if these two are different things, please give a simple overview of them.

6.      In Table. 4, is the multi-sensor monitoring data for tunnel fire conditions used in this article simulated by the author? If yes, what is the approximate amount of simulated data, and whether it is enough?

7.      Could you add more explanation about Figure 8?

8.      To put the current results into context and to enrich readers with the field, the following papers related to the algorithms are to be added to the references, such as Nano-Micro Lett. 16, 214 (2024), Sensors and Actuators B: Chemical, 2024: 136198, Sensors and Actuators B: Chemical 2024, 417, 136206.

Comments on the Quality of English Language

No further comments

Author Response

Dear Reviewers:

Thank you for your comments concerning on our manuscript entitled “Tunnel Fire Detection Method Based on Improved DS Evidence Theory”. (ID: sensors-3156128). Those comments are all valuable and very helpful for revising and improving our paper, as well as the important guiding significance to our researches. We have studied comments carefully and have made correction which we hope meet with approval. The revised parts of the original manuscript were marked in red and the added research parts and simulations according to the reviewer’s comments were marked in blue. The main corrections in the paper and the responds to the comments are as following:

Reviewer 2 Report

Comments and Suggestions for Authors

The paper discusses a tunnel fire detection method based on an improved Dempster-Shafer (DS) evidence theory. It addresses the issue of conflicting data from multi-sensors by proposing a two-level data fusion framework. The experimental results, based on fire scenarios simulated under different conditions, demonstrate the effectiveness of the proposed method in improving both the accuracy and speed of fire detection.

However, there are some problems in the current paper.

1.The abstract mentions that the probability of fire occurrence in four scenarios by this method is 67.5%, 71.0%, 82.8% and 83.5% respectively. This statement is inaccurate. The numerical simulation is of the scenario of fire occurrence, so it should be stated that there is a 67.5%, 71.0%, 82.8% and 83.5% probability of detecting fire when it occurs, rather than the probability of fire occurrence.

2. The paper does not discuss the impact of sensor position on the detection results, which needs to be considered.

3. It is recommended to use higher wind speeds or different fire source locations to test the robustness of the method.

4. The paper does not analyze why the detection accuracy of the proposed method under the four conditions varies greatly. Is it possible that the accuracy will decrease in other scenarios?

Comments on the Quality of English Language

N/A

Author Response

(The authors gave the same response as above.)
